# Chromosome-specific maturation of the epigenome in the *Drosophila* male germline

**James T Anderson[1], Steven Henikoff[1,2]\*, Kami Ahmad[1]\***

[1]Basic Sciences Division, Fred Hutchinson Cancer Center, Seattle, United States;
[2]Howard Hughes Medical Institute, Chevy Chase, United States

**\*For correspondence:**
steveh@fhcrc.org (SH);
kahmad@fredhutch.org (KA)

**Competing interest:** The authors declare that no competing interests exist.

**Abstract** Spermatogenesis in the *Drosophila* male germline proceeds through a unique transcriptional program controlled both by germline-specific transcription factors and by testis-specific versions of core transcriptional machinery. This program includes the activation of genes on the heterochromatic *Y* chromosome, and reduced transcription from the *X* chromosome, but how expression from these sex chromosomes is regulated has not been defined. To resolve this, we profiled active chromatin features in the testes from wildtype and meiotic arrest mutants and integrate this with single-cell gene expression data from the Fly Cell Atlas. These data assign the timing of promoter activation for genes with germline-enriched expression throughout spermatogenesis, and general alterations of promoter regulation in germline cells. By profiling both active RNA polymerase II and histone modifications in isolated spermatocytes, we detail widespread patterns associated with regulation of the sex chromosomes. Our results demonstrate that the *X* chromosome is not enriched for silencing histone modifications, implying that sex chromosome inactivation does not occur in the *Drosophila* male germline. Instead, a lack of dosage compensation in spermatocytes accounts for the reduced expression from this chromosome. Finally, profiling uncovers dramatic ubiquitinylation of histone H2A and lysine-16 acetylation of histone H4 across the *Y* chromosome in spermatocytes that may contribute to the activation of this heterochromatic chromosome.

## eLife assessment

Using a variety of methods including mutant analyses, the authors study chromatin structure during spermatogenesis in *Drosophila* and transcriptional profiling in single cells/nuclei. This description of the dramatic changes in chromatin structure during spermatogenesis leads to some new observations, with **convincing** evidence, and it is **useful** for the field.

## Introduction

The germline in animals is responsible for producing the specialized gametes that transmit genetic information to the next generation, and gene expression in these cells is tightly controlled to ensure proper cell differentiation and genome stability. In many cases gene regulation is distinctive from that in somatic cells (*Freiman, 2009*). A dramatic example of this is the deployment of testis-specific variants of core transcriptional machinery in the *Drosophila* male germline that are used to activate and regulate gene promoters during sperm development (*Hiller et al., 2004*). These variants enable the activation of testis-specific promoters, but the effects they have on transcription and chromatin are less characterized.

Gene regulation in *Drosophila* spermatogenesis also involves widespread changes on entire chromosomes. The sex chromosomes each have distinct chromosomal features which uniquely impact their

expression in the germline: the single *X* chromosome of males is thought to be up-regulated in early germline cells, but then suffers a broad reduction in expression during spermatogenesis (*Witt et al., 2021*; *Mahadevaraju et al., 2021*). In contrast, the *Y* chromosome is largely heterochromatic and silenced in somatic cells but becomes highly active in spermatocytes, extruding long, diffuse loops of actively transcribing genes within the nucleus (*Fingerhut et al., 2019*). How these chromosome-wide changes are orchestrated remains unknown.

Here, we apply CUT&Tag chromatin profiling (*Kaya-Okur et al., 2019*) on the adult testis of *Drosophila* to characterize the chromatin features of the spermatogenic transcription program. By integrating chromatin profiles with published single-cell transcriptional data for the *Drosophila* testis, we track active chromatin features of germline-specific gene promoters, detailing their timing of activation. Further, we describe the enrichment of RNA polymerase II and select histone modifications across the sex chromosomes in isolated spermatocytes. These profiles show that the *X* chromosome does not accumulate repressive chromatin marks, supporting a model where reduced expression of this chromosome is due to a lack of chromosomal dosage compensation in this cell type. Surprisingly, we find that high levels of mono-ubiquitinylated histone H2A accumulate across the *Y* chromosome in spermatocytes, implicating this otherwise repressive histone modification in the modulation of heterochromatic regions in the male germline.

## Results
### Profiling active promoters in the *Drosophila* testes
To profile chromatin features of gene activity during spermatogenesis, we performed CUT&Tag profiling for histone H3 lysine-4 dimethylation (H3K4me2), which marks active promoters and enhancers (*Bernstein et al., 2005*). The adult testis contains all developmental stages of spermatogenesis, as cells in the germline continually proliferate and differentiate (*Figure 1A*; *White-Cooper and Bausek, 2010*). Germline stem cells at the apical tip of the testis asymmetrically divide to birth a gonialblast; these undergo four mitotic divisions and then complete one final S phase before entering an extended G2 phase as primary spermatocytes before meiosis and differentiation into sperm. The transcriptional program of pre-meiotic and post-meiotic stages has been detailed most extensively by single-cell RNA-seq profiling (*Witt et al., 2019*; *Shi et al., 2020*; *Witt et al., 2021*; *Mahadevaraju et al., 2021*; *Raz et al., 2023*). To assess the developmental timing of active chromatin features, we profiled the H3K4me2 modification in wildtype adult testes and in two stage-arrest mutants. The germline of *bag-of-marbles* (*bam*) mutant males arrests in the spermatogonial stage, so their testes are full of early germline cells (*Chen et al., 2011*). The germline of *always early* (*aly*) mutants arrests in the early primary spermatocyte stage, thus enriching for this cell type (*Laktionov et al., 2018*). Between these three genotypes, only wildtype testes contain late spermatocytes and post-meiotic stages. Thus, profiles of testes from these flies distinguishes when in development active chromatin features appear. We dissected testes from 1-day-old adult wildtype, *bam*, and *aly* males, sequenced three replicate libraries for each genotype, and mapped reads to a repeat-masked version of the *Drosophila* dm6 genome assembly (see Methods). To aid distinguishing germline from somatic cell-type chromatin features, we also profiled wing imaginal discs from larvae. Sequencing from replicates was pooled, providing 4.4–19.2M reads/genotype, and publicly available coverage tracks are posted here.

Inspection of these tracks reveals active chromatin features that track with gene expression timing during spermatogenesis. For example, the promoter for the germline stem cell marker *nanos* (*nos*) is marked with the H3K4me2 modification in *bam* testes, with less signal in *aly* or in wildtype testes (*Figure 1B*). In contrast, the promoter of the spermatocyte-expressed gene *loopin-1* lacks the H3K4me2 modification in *bam* testes but is heavily marked in *aly* and in wildtype testes (*Figure 1C*). Similarly, the promoter of the meiotically expressed genes *Mst36Fa* and *Mst36Fb* (*Di Cara et al., 2006*) are marked with the H3K4me2 modification in wildtype testes where the later stages of spermatogenesis are present (*Figure 1D*). In contrast, no signal is present at the *nos*, *loopin-1*, *Mst36Fa*, or *Mst36Fb* genes in wing discs (*Figure 1B–D*). Finally, profiles for all three testes samples show the H3K4me2 mark at the promoters of the homeotic genes *abdominal-A* and *Abdominal-B* that are expressed in the somatic cells of the testis (*Figure 1E*), since somatic cells are present in all three samples.

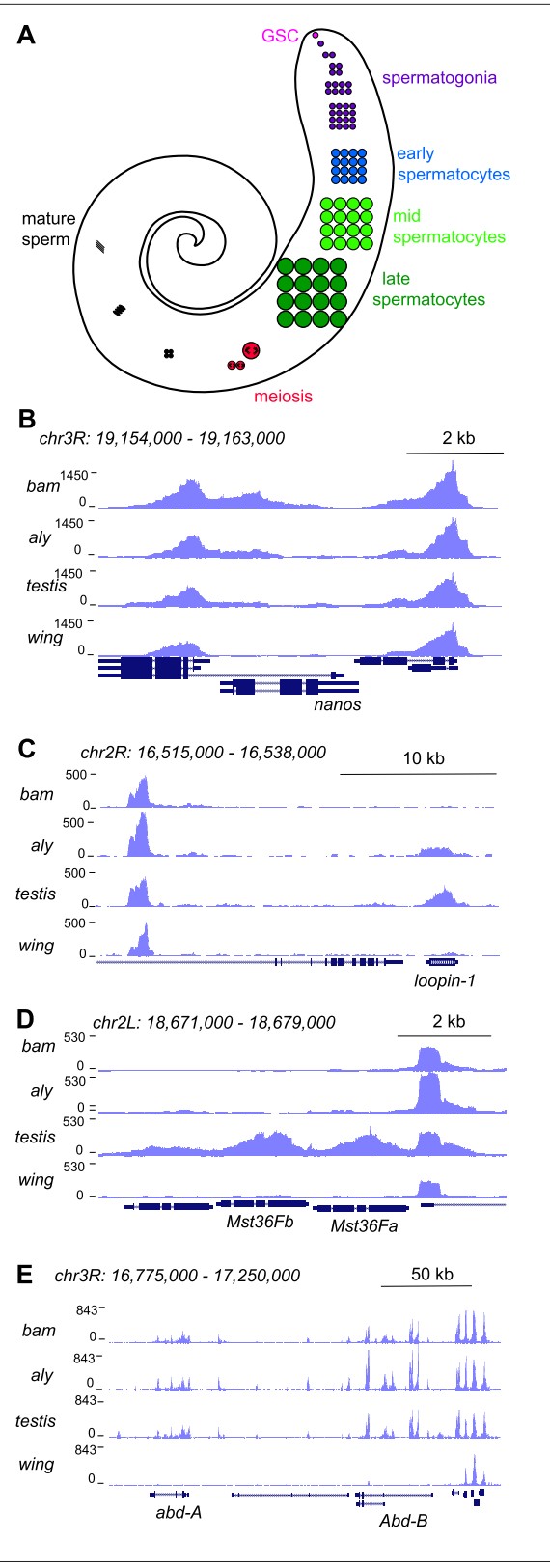

**Figure 1.** Profiling of the histone H3K4me2 modification in the *Drosophila* testis. (**A**) Schematic of male germline stages in *Drosophila*. Germline stem cells (GSCs, fuschia) are located in the apical tip of the testis. After an asymmetric division a progeny spermatogonium (purple) undergoes four rounds of mitotic divisions. After one last S phase cells grow over ~3 days as spermatocytes (blue, light green, dark green) before meiosis (red). Post-meiotic

*Figure 1 continued on next page*

*Figure 1 continued*

differentiation produces mature sperm (black) with elongated nuclei. Somatic cell types of the testis are not shown. (**B**–**E**) Distribution of the H3K4me2 modification in testes from *bam* mutants, from *aly* mutants, from wildtype animals, and from wing imaginal discs. (**B**) H3K4me2 around the GSC-expressed *nanos* gene. Neighboring genes show peaks in all samples, while low signal across *nanos* is highest in testes from *bam* mutants, and apparent in all three testes samples. (**C**) H3K4me2 around the spermatocyte-expressed *loopin-1* gene. H3K4me2 signal appears in *aly* mutant samples (which contain early spermatocytes) and reach high levels in wildtype testes (which include later stages of spermatogenesis). (**D**) H3K4me2 around the meiotically expressed genes *Mst36Fa* and *Mst36Fb* genes. Signal across these genes only appears in wildtype testes. (**E**) H3K4me2 around the *abd-A* and *Abd-B* genes, which are expressed in somatic cells of the testis.

To visualize chromatin features across active promoters during spermatogenesis, we categorized genes by their timing and level of mRNA expression in the male germline in single-nucleus RNA-seq profiling (*Figure 2*; *Raz et al., 2023*). We focused on five categories of genes with germline-enriched expression in germline stages (*Figure 2—figure supplement 1*, *Supplementary file 1b and c*), including 845 genes that are predominantly expressed in spermatogonia, 1510 expressed in early spermatocytes, 1524 expressed in mid-spermatocytes, 2052 expressed in late spermatocytes, and 475 genes expressed in spermatids. We then displayed the summed H3K4me2 signal spanning –200 to +500 bp around each of the promoters for these genes (*Figure 2A*). In the whole testis, germline stages compose a small proportion of all cells, and so genes specifically expressed in these stages have low signal for both mRNA and the H3K4me2 modification (*Supplementary file 1b*). Nevertheless, the overall tendency is that promoters of genes expressed in spermatogonia are marked with the H3K4me2 modification in *bam* mutant testes, which are enriched for early stages compared to *aly* mutant and wildtype testes (*Figure 2A*). For example, the promoters of early germline markers *nos*, *vasa* (*vas*), and *zero population growth* (*zpg*) are each heavily marked with H3K4me2 in *bam* mutant samples, reflecting their activity (*Figure 2A, B*). Similarly, the promoters for many genes primarily expressed in early spermatocytes and mid-stage spermatocytes are often most heavily marked with H3K4me2 in bam mutant testes, implying that these genes first become active in spermatogonia and accumulate mRNA in spermatocyte stages. Finally, gene promoters for mRNA that accumulate in late spermatocytes and in differentiating spermatids are predominantly marked with the H3K4me2 modification in wildtype testes, as this is the only sample that contains these stages of spermatogenesis (*Figure 2A*). This includes the activation of promoters for the *Y* chromosome-linked fertility factor genes (*kl-2*, *kl-3*, and *kl-5*) which are primarily expressed in late spermatocytes (*Figure 2A, B*). However, the activation timing of these genes appears to differ, as the promoters of *kl-2* and *kl-3* accumulate the H3K4me2 modification earlier than does the *kl-5* promoter (*Figure 2B*). The most dramatic instance of precocious activation of a *Y*-linked promoter is the *FDY* gene which becomes active in early spermatocytes, matching its early production of mRNA (*Figure 2A, B*).

There are three exceptions to the overall trend of correspondence between promoter activation and mRNA accumulation. First, active promoters with very low levels of H3K4me2 are most heavily marked in wildtype testes, regardless of when the genes are expressed. These might be active genes where the histone modification accumulates during the extended growth phase of spermatocytes. Additional examples are the *kumgang* (*kmg*) and *cookie monster* (*comr*) gene promoters which produce mRNA in mid-stage spermatocytes, but are most heavily marked with the H3K4me2 modification in wildtype testes. Second, some promoters acquire the H3K4me2 modification well before mRNA accumulates. A small number of genes are expressed in the post-meiotic stages of spermatogenesis, including the 'cup' genes (*Barreau et al., 2008*). Many of these promoters are most heavily marked with the H3K4me2 modification in wildtype testes, but the promoter for *ryder cup* (*r-cup*) is already marked in *aly* mutant testes, implying that it is already active in pre-meiotic stages (*Figure 2B*). Third, as the arrest mutations used here delete most of this gene (*bam*, *Bopp et al., 1993*) or inactivate it (*aly*, *Lin et al., 1996*), we cannot measure H3K4me2 modification at these promoters in their arrest genotypes. We note that as these stage-arrest mutations have pleiotropic effects on gene expression (*Barreau et al., 2008*), some discrepancies between promoter marking in mutants and transcript accumulation in wildtype testes may be due to aberrant transcriptional regulation. Additionally, detection of some changes may be limited since we compare profiles of tissues with diverse cell types.

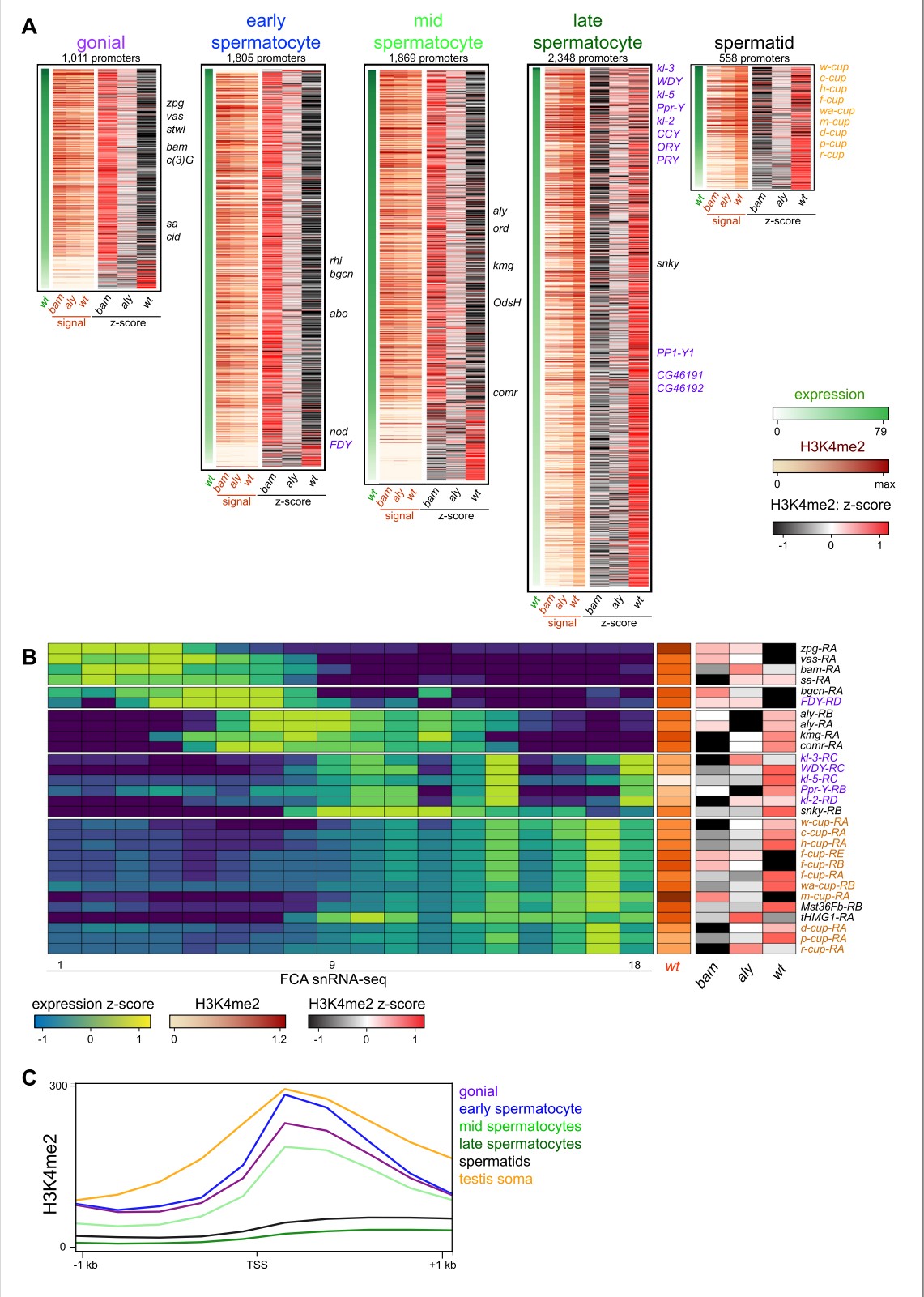

**Figure 2.** Changes in the histone H3K4me2 modification in germline-expressed genes. H3K4me2 signal around gene promoters (–200 to +500 bp) with transcripts enriched in specific germline cell types. Transcript expression was derived from FCA snRNA-seq clustering (*Raz et al., 2023*). (**A**) Expression (snRNA-seq) in wildtype testes and H3K4me2 enrichment in *bam* mutant, in *aly* mutant, and in wildtype testes in germline stages. Z-scores for H3K4me2 signal were calculated between the three genotypes. Notable germline-enriched genes are indicated, including those expressed in GSCs, in meiosis, or

*Figure 2 continued on next page*

*Figure 2 continued*

for cell division (black), linked to the *Y* chromosome (purple), or expressed post-meiotically (orange). (**B**) Selected examples of promoters with germline-enriched expression. Expression z-scores from FCA snRNA-seq across 18 germline clusters (**Raz et al., 2023**) and H3K4me2 enrichment in *bam* mutant, in *aly* mutant, and in wildtype testes. (**C**) Distribution of H3K4me2 around promoters with germline-enriched expression. The testis somatic category comprises the top tercile of promoters with somatic cell-type expression in snRNA-seq data (**Raz et al., 2023**). Only promoters with no promoter of a second gene within 1 kb upstream are shown.

The online version of this article includes the following figure supplement(s) for figure 2:

**Figure supplement 1.** Genes with germline-enriched expression in testes.

Notably, the importance of the H3K4me2 modification appears to diminish as spermatogenesis proceeds, as the average signal of this mark around promoters for germline-enriched genes in wild-type testes drops in the later stages of spermatogenesis (*Figure 2C*). While the promoters of early germline-expressed genes and somatically expressed genes have comparable levels of the H3K4me2 modification centered around their TSS, the promoters of late spermatocyte and spermatid genes display very little marking, and this low level extends into gene bodies. This suggests that the activities of H3K4-modifying enzymes are reduced in these later stages.

## Profiling FACS-isolated primary spermatocytes

Germline cells undergo massive nuclear expansion and extensive transcriptional activation between mitotic spermatogonia and meiotic division, in part directed by germline-specific variants of general transcription factors (*Lim et al., 2012*). To specifically profile chromatin features in spermatocytes, we used a spermatocyte-enriched GFP marker to isolate these cells by fluorescence-activated cell sorting (FACS). The *hephaestus* (*heph*) gene encodes an RNA-binding protein that is broadly expressed, but in spermatocytes Heph binds the abundant nuclear transcripts from the *Y* chromosome fertility genes (*Figure 3A*; *Fingerhut et al., 2019*). We performed FACS on 40 dissociated testes from males carrying a *heph-GFP* transgene, recording the forward scatter (FSC) and GFP signal of each event (*Figure 3B*). FACS profiles from *heph-GFP* samples display a large proportion of events with high GFP signal, which are absent in profiles of wildtype testes. These GFP-labeled spermatocytes have distinct sizes, consistent with the progressive growth of spermatocytes as they approach meiosis (*White-Cooper et al., 2010*): in a typical FACS experiment, ~5–10% of GFP-positive events have moderate GFP signal and moderate size (Gate 1), ~50% of events have very high GFP signal and moderate size (Gate 2), and ~5–10% have high GFP signal and large size (Gate 3). The cells of Gate 3 may represent the latest stage of spermatocytes when heph-GFP signal decreases just before meiosis. Because Gate 2 contained the most GFP-positive events, we focused further analysis on these spermatocytes. We used ~3000 isolated spermatocytes for each profiling experiment, and since the resulting libraries were comparatively small with high duplication rates, we pooled unique reads from multiple replicates to provide 200,000–900,000 unique reads for each profile (*Supplementary file 1a*).

We first profiled the distribution of the elongating form of RNA Polymerase II, marked with phosphorylation at Serine-2 (RNAPIIS2p) of the C-terminal tail of the largest subunit of the complex. Inspection of genome landscapes demonstrates the high quality of these profiles. For example, the meiotic beta-tubulin variant gene *betaTub85D* is broadly coated with RNAPIIS2p in isolated spermatocytes, while signal is absent across this gene in somatic cells (*Figure 3C*).

Elongating RNAPII is also detectable at many genes that accumulate transcripts in late spermatocytes. For example, 14 genes encoding protamines that package the genome in sperm (*Chang et al., 2023*) are heavily coated with RNAPII in isolated spermatocytes (*Figure 3E*). Similarly, elongating RNAPII is detectable at genes normally thought to be expressed in post-meiotic cells, such as *heineken-cup* (*h-cup*), implying that RNAPII is engaged at some genes well before their transcripts are detected.

While these results confirm the cell-type identity of the FACS-isolated cells, we noted that the distribution of RNAPIIS2p across genes such as *betaTub85D* differs from the typical pattern across active genes in somatic cells. Serine-2-phosphorylation of RNAPII is associated with transcriptional elongation, but in *Drosophila* and in mammalian somatic cells it shows a prominent peak at the 5' end of active genes (*Kaya-Okur et al., 2019*; *Ahmad and Henikoff, 2021*). The broad distribution of elongating RNAPII across active genes is typical in spermatocytes. Genes with germline-enriched

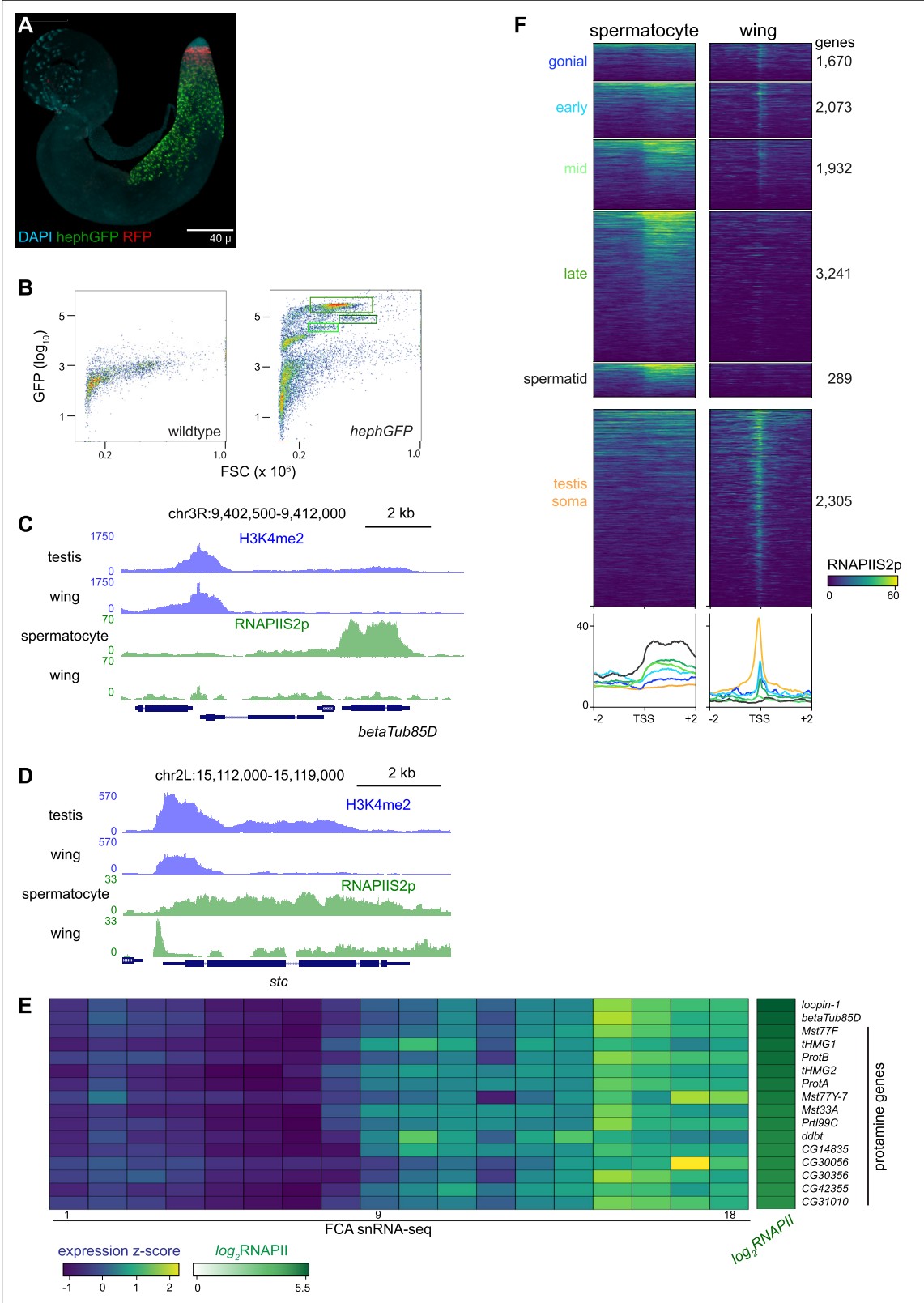

**Figure 3.** Profiling RNA polymerase II in isolated spermatocytes. (**A**) An adult testis carrying a *UASRFP* construct induced by a *bamGAL4* driver and a *hephGFP* construct. Gonial cells are labeled red while spermatocytes are labeled green with fluorescent proteins. (**B**) Fluorescence-activated cell sorting (FACS) plots of recorded events from dissociated testes for forward scatter (FSC) and GFP signal. Boxes indicate events collected for chromatin profiling. (**C**) Distribution of RNAPIIS2p at the spermatocyte-expressed *betaTub85D* gene in isolated spermatocytes and in wing imaginal discs. (**D**)

*Figure 3 continued on next page*

*Figure 3 continued*

Distribution of RNAPIIS2p at the broadly expressed *stc* gene in isolated spermatocytes and in wing imaginal discs. RNAPIIS2p is strongly localized at the *stc* promoter in wing imaginal discs, but more evenly distributed in spermatocytes. (**E**) Selected examples of genes with late germline expression and for protamines. Expression z-scores from FCA snRNA-seq across 18 germline clusters (**Raz et al., 2023**) and RNAPIIS2p enrichment in isolated spermatocytes. (**F**) Enrichment of RNAPIIS2p in isolated spermatocytes and in wing imaginal discs across genes with germline-enriched transcripts.

The online version of this article includes the following figure supplement(s) for figure 3:

**Figure supplement 1.** Distribution of elongating RNAPII in spermatocytes and in somatic cells.

expression show a broad distribution of elongating RNAPII downstream of their promoters, while somatically expressed genes show a prominent 5′ peak (**Figure 3F**).

To clearly compare RNAPII distributions between cell types, we examined long genes that are commonly expressed in both spermatocytes and in wing imaginal discs. One example is the *shuttle craft* (*stc*) gene, which is almost 5 kb and is highly expressed. Strikingly, RNAPIIS2p signal at *stc* shows a prominent peak near the promoter in somatic wing imaginal disc cells but is broadly distributed across the gene in spermatocytes (**Figure 3D**). More generally, RNAPIIS2p is broadly distributed across all active genes in spermatocytes, in contrast to the proximally peaked distribution in somatic cells (**Figure 3—figure supplement 1**).

The change from peaked to broad distributions of RNAPIIS2p is mirrored in the distributions of the H3K4me2 modification, as this histone modification shows an atypical broad and low distribution across the active *betaTub85D* and *stc* genes in spermatocytes (**Figure 3C, D**). As this histone modification occurs co-transcriptionally, its change in distribution is likely the result of the altered distribution of RNAPII across these genes.

## The *X* chromosome is not dosage compensated in spermatocytes

In somatic cells of *Drosophila* males, the expression of genes on the single *X* chromosome is approximately doubled to equalize expression to autosomal genes. Canonical dosage compensation is accomplished by the male-specific lethal RNA-protein complex, which coats the *X* chromosome, catalyzes acetylation of histone H4 at lysine 16 (H4K16ac), and increases RNAPII density (**Akhtar and Becker, 2000**). However, in germline cells in the testis cytology detects no enrichment of the H4K16ac modification, suggesting that *X* chromosome dosage compensation does not occur in this cell type (**Rastelli and Kuroda, 1998**). Transcriptomic profiling showed that multiple components of the dosage compensation machinery are not expressed in the male germline (**Witt et al., 2021**). In spite of this, single-cell RNA-seq studies have shown that a perhaps non-canonical form of *X* chromosome dosage compensation occurs in early spermatogonial stages, but disappears by spermatocyte stages (**Mahadevaraju et al., 2021**; **Raz et al., 2023**; **Witt et al., 2021**).

The distribution of RNAPII in isolated spermatocytes is consistent with the lack of dosage compensation by this stage of spermatogenesis. Plotting the distribution of RNAPIIS2p across *Drosophila* chromosomes in wing imaginal discs shows a substantial enrichment across the *X* chromosome, resulting from dosage compensation in these somatic cells (**Figure 4A**). This enrichment of RNAPIIS2p across the *X* chromosome is lost in spermatocytes. To quantify chromosomal changes, we summarized RNAPIIS2p signal for the autosomal second and third chromosomes, the quasi-heterochromatic fourth chromosome, and the sex chromosomes (**Figure 4B**). As the sex chromosomes are hemizygous, we doubled counts for genes on these chromosomes to calculate polymerase densities per gene copy, and then scaled gene scores to the median value of gene scores on the second and third autosomal chromosomes. As expected, the median expression of *X*-linked genes in wing imaginal disc cells is close to twice that of the major autosomes, showing they are dosage compensated. In contrast, median expression from *X*-linked genes is equal to that of the major autosomes in spermatocytes (**Figure 4B**).

To assess the chromosomal distribution of the H4K16ac modification, we profiled it in wing imaginal discs and in isolated spermatocytes. This acetylation is widespread across the genome, consistent with its association with transcriptional activity, and in wing imaginal disc cells it is noticeably enriched across the dosage-compensated *X* chromosome (**Figure 4C**). In stark contrast, the *X* chromosome is depleted for H4K16ac in spermatocytes. Thus, chromatin profiling for both elongating RNAPII and the H4K16ac modification demonstrates there is no dosage compensation of the *X* chromosome in spermatocytes.

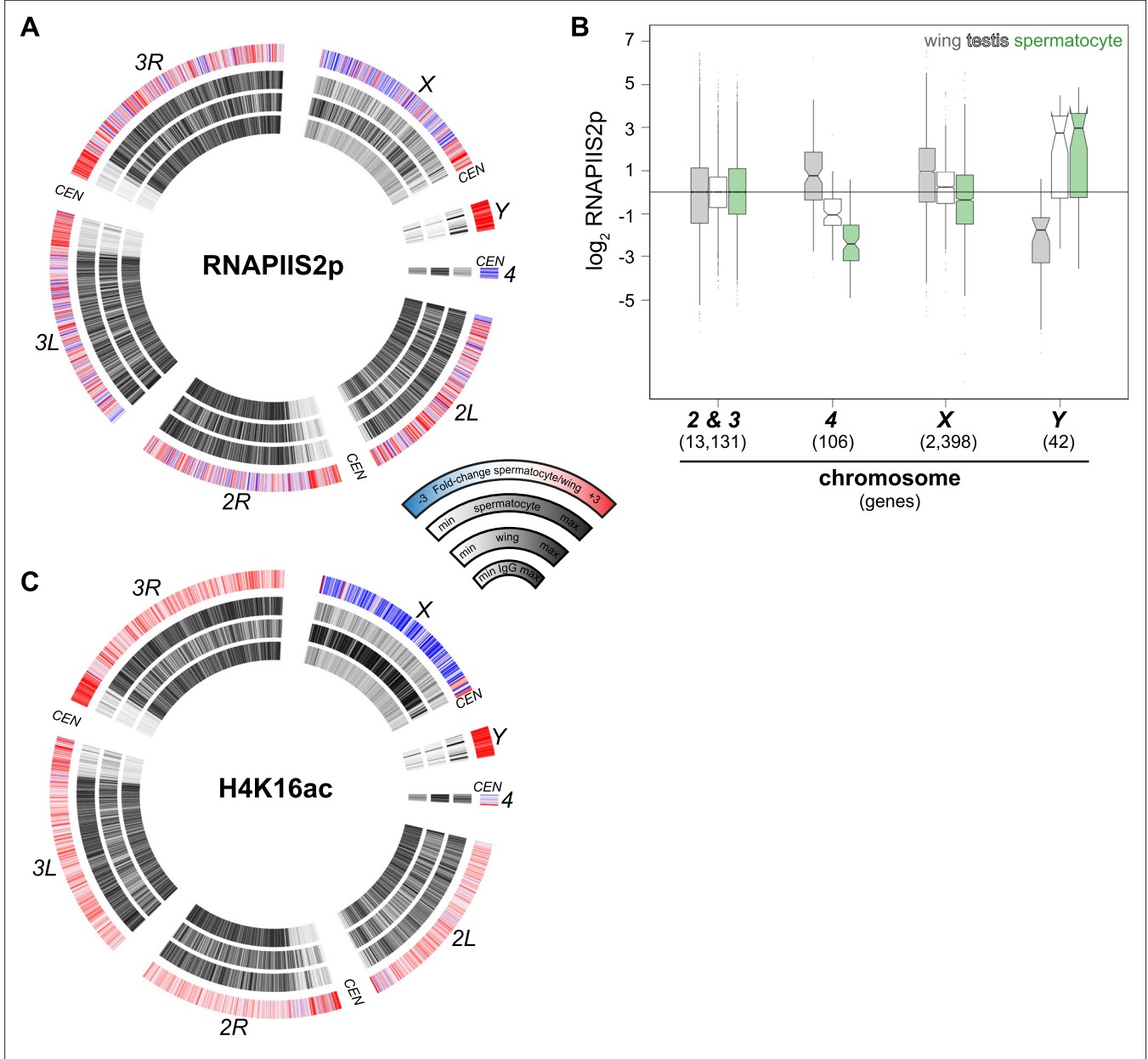

**Figure 4.** Chromosomal distribution of RNA polymerase II and H4K16ac in isolated spermatocytes. (**A**) CIRCOS plot of RNAPIIS2p across *Drosophila* chromosomes. The signal (black) in IgG controls, in wing imaginal discs, and in isolated spermatocytes is shown in internal rings, and the log2fold-change of signals between spermatocytes and wings is shown in the outer ring. (**B**) Enrichment of RNAPIIS2p across gene bodies in wing imaginal discs, in whole dissociated testes, and in isolated spermatocytes separated by chromosomal location. Scores are scaled to the median score on the second and third chromosomes. (**C**) CIRCOS plot of the dosage-compensation marker histone H4K16ac across *Drosophila* chromosomes. Signal (black) in IgG controls, in wing imaginal discs, and in isolated spermatocytes is shown in internal rings, and the log2fold-change of signals between spermatocytes and wings is shown in the outer ring.

RNAPII density on the quasi-heterochromatic fourth chromosome is reduced in spermatocytes, consistent with decreased transcript production from this chromosome in RNA-seq studies (*Mahade-varaju et al., 2021*; *Witt et al., 2021*; *Raz et al., 2023*). The fourth chromosome is an evolutionary derivative of the X chromosome and it has been speculated that it may be subject to similar chromosomal regulation as the X (*Larsson and Meller, 2006*); however, only 10 genes are expressed from this small chromosome in spermatocytes, and this limits any inference about down-regulation of this chromosome. In contrast, the specific activation and accumulation of RNAPIIS2p on Y chromosome genes in spermatocytes is dramatic (*Figure 4A, B*). Likewise, the Y chromosome becomes conspicuously

enriched for the H4K16ac modification in spermatocytes (*Figure 4C*), suggesting this modification is involved in gene activation from this chromosome.

## Profiling silencing chromatin marks in spermatocytes

The sex chromosomes of male therian mammals form a cytological sex body in pre-meiotic cells (*Solari, 1974*). This body is a manifestation of meiotic sex chromosome inactivation (MSCI), where repressive histone modifications silence unpaired chromosomes (*Turner, 2015*). MSCI has been suggested to occur in the *Drosophila* male germline to explain the mysterious dominant male sterility of many *X*-to-autosome translocations (*Lifschytz and Lindsley, 1972*). However, transcriptional profiling of the *Drosophila* testis has not observed silencing of the *X* chromosome (*Mahadevaraju et al., 2021*; *Witt et al., 2021*; *Raz et al., 2023*). We therefore profiled silencing histone modifications in isolated spermatocytes to determine if molecular marks of MSCI are enriched on the *Drosophila X* chromosome. Methylation of histone H3 at lysine-9 (H3K9me) is generally associated with heterochromatic silencing and marks the precociously silenced *X* chromosome in male mouse spermatogenesis (*Khalil et al., 2004*; *Ernst et al., 2019*). In wing imaginal disc cells dimethylation of H3K9 (H3K9me2) is enriched in silenced pericentromeric regions of all chromosomes, as well as throughout the heterochromatic *Y* chromosome and the quasi-heterochromatic fourth chromosome (*Figure 5A*), consistent with the silencing of repetitive sequence regions in somatic cells. However, the genome in spermatocytes gains the H3K9me2 modification throughout chromosome arms, including those of the major autosomes and the *X* chromosome. In contrast, the H3K9me2 modification is reduced across the *Y* chromosome, consistent with the activation of *Y*-linked genes in this cell type (*Figure 5A*). Although the H3K9me2 mark is reduced across this chromosome and across pericentromeric regions, substantial chromosomal methylation remains in spermatocytes.

A major system of chromatin repression uses trimethylation of histone H3 at lysine-27 (H3K27me3) to direct developmental gene silencing (*Grossniklaus and Paro, 2014*). Although this mark is not associated with MSCI in mammals (*Mu et al., 2014*), we profiled it in *Drosophila* spermatocytes. There is little change in the chromosomal distribution of the H3K27me3 modification between wing imaginal disc cells and spermatocytes (*Figure 5B*). There is a slight apparent reduction of this modification across the *X* chromosome and a slight gain across the *Y* chromosome, but these differences may be due to the developmental lineages of these two samples. Overall, the constancy of chromosomal patterns of the H3K27me3 modification is consistent with the inactivation of the histone methyltransferase *Enhancer of zeste* (*E(z)*) in spermatocytes (*Chen et al., 2011*).

## Mono-ubiquitinylation of histone H2A marks the activated *Y* chromosome

An additional histone modification associated with precocious silencing of the *X* chromosome in male mammals is the mono-ubiquitinylation of histone H2A at lysine-119 (uH2A) (*Baarends et al., 1999*). This modification is conserved at the homologous lysine-118 position of *Drosophila* histone H2A, and is linked to Polycomb-mediated silencing across eukaryotes (*Barbour et al., 2020*). We therefore profiled the distribution of the uH2A modification in wing imaginal discs and in isolated spermatocytes to determine if this modification marked the *X* chromosome. The uH2A mark is broadly enriched throughout the arms of autosomes in both cell types, but shows no enrichment across the *X* chromosome in spermatocytes (*Figure 6A*). Thus, this chromatin marker of mammalian MSCI is also absent from the *Drosophila X* chromosome. However, the active *Y* chromosome is strikingly enriched for the uH2A modification in spermatocytes, with additional moderate enrichment in the repetitive pericentromeric regions of all chromosomes (*Figure 6A*).

We confirmed the chromosomal enrichment of the uH2A modification by immunostaining spermatocytes (*Figure 6B*). The subnuclear pattern of a Polycomb-GFP fusion protein distinguishes early from late spermatocytes (*Dietzel et al., 1999*; *El-Sharnouby et al., 2013*). Using this marker, we see that uH2A is largely absent from the nucleus of early spermatocytes. In mid-stage spermatocytes a stringy wedge of uH2A staining appears in the interchromosomal space between the chromatin bodies and expands to one or two wedges in late spermatocytes (*Figure 6B*). The timing of appearance and position of these stained bodies resemble that of the chromosome loops that unfold from *Y*-linked genes as they are expressed (*Bonaccorsi et al., 1988*). We therefore engineered *XO* male flies lacking a *Y* chromosome and immunostained their spermatocytes. The uH2A modification

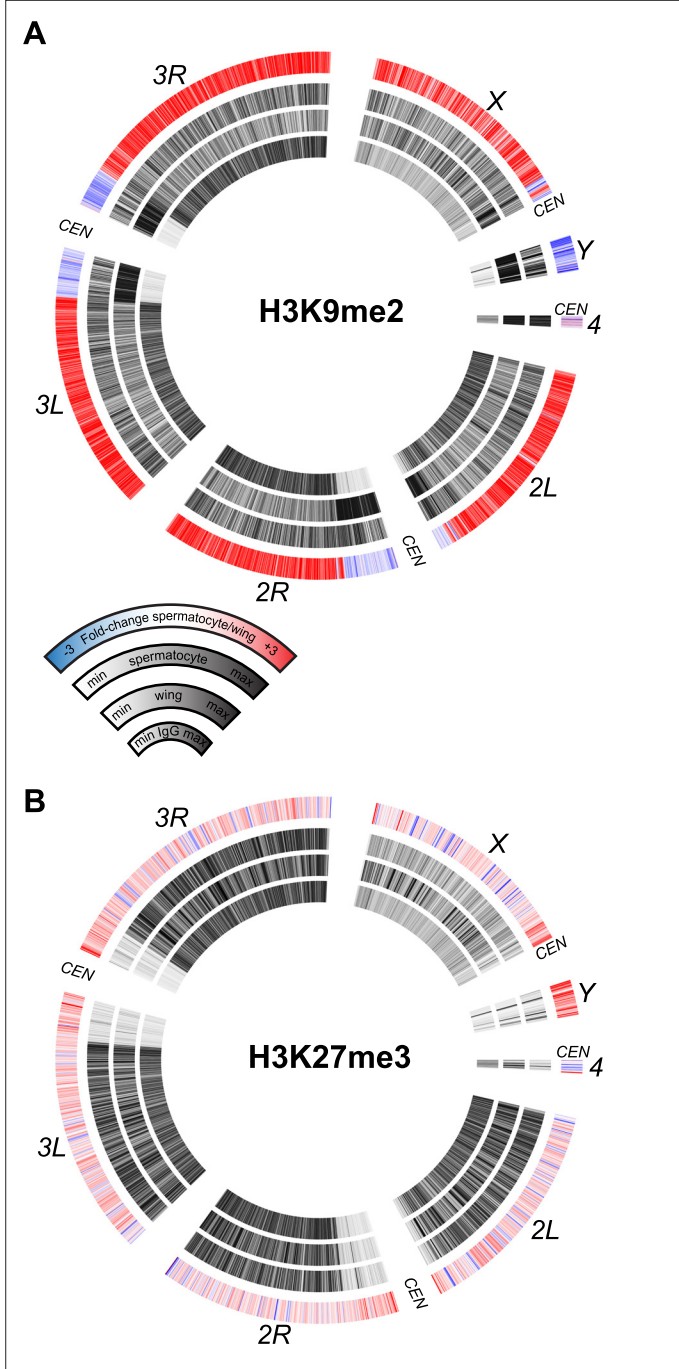

**Figure 5.** Chromosomal distribution of repressive histone modifications in isolated spermatocytes. CIRCOS plots across *Drosophila* chromosomes show signal (black) in IgG controls, in wing imaginal discs, and in isolated spermatocytes in internal rings, and the log$_2$fold-change of signals between spermatocytes and wings in the outer ring. (**A**) Distribution of the heterochromatin-silencing marker H3K9me2. (**B**) Distribution of the Polycomb-silencing marker H3K27me3.

is present in these cells, but with a distinctly different focal appearance, suggesting that the ubiquit-inylated histone aggregates in spermatocytes without a *Y* chromosome (*Figure 6B*). To confirm the timing of the appearance of the uH2A body, we immunostained germline nuclei from *bam* and *aly* stage-arrest mutants. The spermatogonial nuclei from *bam* mutant testes lack uH2A staining, while early spermatocyte nuclei from *aly* mutant testes have only a small uH2A body that always abuts against the nucleolus (*Figure 6B*). Thus, the uH2A body accumulates during the early spermatocyte

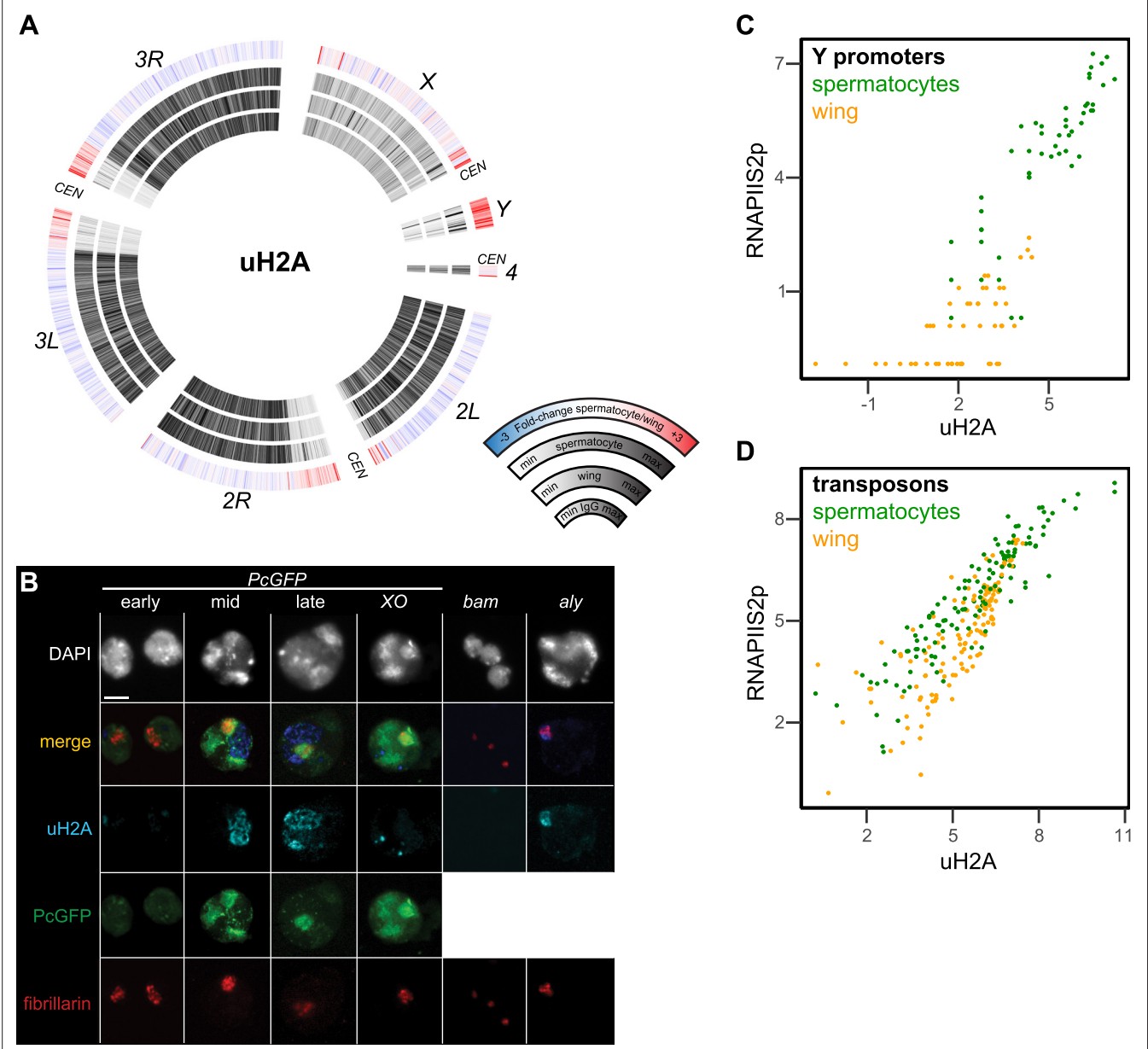

**Figure 6.** Chromosomal distribution of ubiquitinylated histone H2A in isolated spermatocytes. (**A**) CIRCOS plots of uH2A across *Drosophila* chromosomes in IgG controls, in wing imaginal discs, and in isolated spermatocytes. (**B**) Immunostaining of uHA (blue) and the nucleolar marker fibrillarin (red) on germline nuclei. Early-, mid-, and late-spermatocyte stages were identified by Polycomb-GFP (PcGFP) (green) localization pattern in wildtype spermatocytes and in *X/O* spermatocytes. Testes from *bam* mutants contain gonial cells, while testes from *aly* mutants contain mostly early spermatocytes. All images are displayed at the same magnification, and a 100 µm scale bar is shown in the top left section. (**C**) Correspondence of uH2A and RNAPIIS2p signals around the promoters of genes on the *Y* chromosome in wing imaginal discs and in isolated spermatocytes. (**D**) Correspondence of uH2A and RNAPIIS2p signals across transposon consensus sequences in wing imaginal discs and in isolated spermatocytes.

The online version of this article includes the following figure supplement(s) for figure 6:

**Figure supplement 1.** Chromatin features across transposon consensus sequences in spermatocytes and in somatic cells.

stage and expands as spermatocytes develop. The timing and position of the uH2A body is consistent with the idea that it contains the *Y* chromosome, which is transcriptionally active in these stages.

To further characterize the relationship between transcriptional activity of the *Y* chromosome and the uH2A modification, we plotted signal for the uH2A modification and for RNAPIIS2p at *Y*-linked gene promoters (*Figure 6C*). In wing imaginal discs these promoters have little RNAPIIS2p or uH2A modification, but there is strong relative enrichment for both features at these promoters in

spermatocytes, consistent with transcriptional activity of the *Y* chromosome. Since the uH2A modification also becomes enriched in pericentromeric regions, we compared the enrichment of the uH2A modification and RNAPIIS2p at repetitive transposons that constitute a large fraction of these regions. A number of transposons gain RNAPIIS2p signal specifically in spermatocytes (*Figure 6D*; *Supplementary file 1d*; *Figure 6—figure supplement 1*), consistent with their transcriptional activation (*Lawlor et al., 2021*). These activated transposons also gain the uH2A modification. These correspondences suggest that the uH2A modification modulates transcriptional activation of heterochromatic regions in spermatocytes.

## Discussion

Germline cells use distinctive variations on transcriptional gene regulation, and studies of *Drosophila* spermatogenesis have detailed many specialized alterations of core general transcription factors that direct expression programs as differentiation proceeds (*Hiller et al., 2004*). However, the chromatin features of gene regulation in spermatogenesis have been less thoroughly characterized, in part because of the complexity of the tissue and limiting numbers of germline cells. We have addressed this by performing efficient CUT&Tag chromatin profiling for both active and repressive chromatin marks in the *Drosophila* testis and in isolated spermatocytes. These profiles reveal several notable features of the epigenome in the differentiating germline. First, integration of chromatin marks with published gene expression data details a general correspondence between gene promoter activation and mRNA production as expected, but for a fraction of genes their promoters activate earlier than expected. Second, genes that are activated late in spermatogenesis tend to have very reduced active chromatin marks at their promoters. Third, while many active genes in somatic cells accumulate RNAPII near their gene starts, such accumulation is absent in spermatocytes. Fourth, integration of chromatin profiling for multiple chromatin marks and profiling of RNAPII demonstrate that the single *X* chromosome is neither dosage compensated nor globally inactivated in spermatocytes. Finally, histone H2A mono-ubiquitinylation appears to have a specialized role in modulating expression from the heterochromatic *Y* chromosome.

### Quirks of gene expression in the male germline

The uniform distribution of RNAPII throughout active genes in male germline cells is strikingly different from the typical accumulation of RNAPII near promoters in somatic cells. Promoter-proximal accumulation results from dynamic pausing of RNAPII before conversion into the productive elongating isoform, and is a major control point for transcriptional regulation in somatic cells (*Muniz et al., 2021*). Thus, the uniformity of RNAPII across expressed genes in spermatocytes suggests that pausing does not occur, necessitating gene regulation solely by transcription factor and RNAPII recruitment. This may allow for a simpler promoter architecture, and indeed spermatogenic gene promoters are distinctively small (*White-Cooper et al., 2010*). Alternatively, RNAPII progression through gene bodies may be slow, altering the steady-state distribution of elongating polymerase. Further, either of these changes in RNAPII behavior would affect chromatin features of active genes. RNAPII binds enzymes that progressively methylate the lysine-4 residue of histone H3, and so active promoters in somatic cells are typically marked with both H3K4-dimethylation and -trimethylation (*Bernstein et al., 2005*). A consequence of changing the rate of polymerase progression in spermatocytes would be reduced methylation of active promoters (which we observe is most severe in late spermatocyte and post-meiotic stages), and thereby reduced reliance of these marks for promoting transcription.

### The *X* chromosome is neither dosage-compensated nor inactivated in spermatocytes

While protein and lncRNA components of the somatic dosage compensation machinery are not produced in the male germline, transcriptional profiling described up-regulation of the single *X* chromosome in early germline stages (*Mahadevaraju et al., 2021*; *Witt et al., 2021*). The mechanism for this non-canonical dosage compensation remains unknown (*Witt et al., 2021*), but by the spermatocyte stage *X*-linked genes are no longer up-regulated. Our profiling of RNAPII confirms that there is no up-regulation of this chromosome in spermatocytes. There has been substantial investigation into whether the *X* is inactivated in spermatocytes, inspired by the regulation of sex chromosomes in

mammals (*Vibranovski, 2014*; *Turner, 2015*) and the moderate depletion of male-germline-expressed genes on the *Drosophila X* chromosome (*Parisi et al., 2003*). In mammalian male germlines the *X* and *Y* chromosomes undergo MSCI, where these chromosomes are precociously silenced just before meiosis (*Turner, 2015*). A variety of chromatin features accumulate across the sex chromosomes at this time, including the enrichment of both H3K9me2 and uH2A histone modifications. However, we find that neither of these histone modifications is enriched on the *X* chromosome in the *Drosophila* male germline. This combined with the equivalent amounts of RNAPII on *X*-linked genes and autosomal ones implies that there is no *X* chromosome inactivation in *Drosophila*.

Why do flies differ from mammals? Mammalian MSCI is considered to be an elaborated response of germline cells to the detection of the unpaired sex chromosomes (*Huynh and Lee, 2005*). However, meiosis in *Drosophila* males is unusual in that all the chromosomes do not synapse nor recombine (*McKee and Handel, 1993*). Logically, the evolutionary loss of synapsis must have required the concomitant loss of an unpaired chromosome response in *Drosophila* males. Indeed, many of the proteins recruited to the mammalian sex body are normally involved in meiotic recombination but have been repurposed for unpaired chromosome inactivation (*Abe et al., 2022*), supporting the idea that MSCI is mechanistically linked to synapsis. Alternatively, mammalian MSCI may be a variant of the X-inactivation system that operates in females for dosage compensation (*Huynh and Lee, 2005*) but since *Drosophila* solved the dosage compensation problem without inactivation, this precluded the evolution of MSCI. These possibilities are not mutually exclusive.

## Activation of the *Y* chromosome

The *Y* chromosome in *Drosophila* is unique in that it is almost entirely composed of repetitive sequence, but also carries some unique genes required for male fertility (*Chang and Larracuente, 2019*). Thus, in somatic cell types this a heterochromatic chromosome, but is heavily transcribed in spermatocytes (*Bonaccorsi et al., 1988*). This activated chromosome accumulates multiple histone modifications (*Hennig and Weyrich, 2013*), some of which we have profiled here. Surprisingly, one of the histone modifications that coats the activated *Y* chromosome is mono-ubiquitinylation of histone H2A. This modification is typically associated with Polycomb-mediated gene silencing in somatic cells (*Aloia et al., 2013*), where it is catalyzed by the Sce/RING1B enzyme subunit of the Polycomb repressive complex 1 (PRC1) (*Gorfinkiel et al., 2004*; *Wang et al., 2004*). However, the Polycomb subunit of PRC1 does not co-localize with the uH2A-coated *Y* chromosome in spermatocytes, suggesting that it is not catalyzed by Sce here. Indeed, uH2A also coats the *X* and *Y* chromosomes in mammalian spermatocytes, where this modification is catalyzed by a distinct enzyme, the UBR2 E3 ubiquitin ligase (*An et al., 2010*). The *Drosophila* genome encodes a homolog of this protein family, as well as many other ubiquitin ligases, some of which target the H2A histone (*Tasaki et al., 2005*). We do not know which enzyme is responsible for the uH2A modification in the *Drosophila* male germline, although some candidates have male sterile phenotypes when mutated (*Rathke et al., 2007*). While it is surprising to find a histone modification conventionally associated with silencing enriched on an activated chromosome (and indeed the reason we profiled it was as a putative MSCI marker), the roles of uH2A in gene regulation are diverse even in somatic cells. The uH2A modification is associated with both silenced and some active genes in developing eye tissue (*Loubiere et al., 2020*), and counters chromatin compaction in the early embryo (*Bonnet et al., 2022*). In addition to the *Y* chromosome, a number of heterochromatic transposons in the *Drosophila* male germline are also activated and accumulate the uH2A modification. Thus, it is conceivable that this modification – perhaps in combination with other marks – works generally to modulate transcription of extremely heterochromatic regions in spermatocytes.

# Methods

**Key resources table**

| Reagent type (species) or resource | Designation | Source or reference | Identifiers | Additional information |
| --- | --- | --- | --- | --- |
| Antibody | Anti-H3-K4-dimethyl (rabbit monoclonal) | Epicypher | Cat. No. 13–0027 RRID: AB_3068541 | CUT&Tag (1:100) |

*Continued on next page*

*Continued*

| Reagent type (species) or resource | Designation | Source or reference | Identifiers | Additional information |
|---|---|---|---|---|
| Antibody | Anti-RNAPII-Serine-2-phosphorylation (rabbit monoclonal) | Cell Signalling Technology | Cat. No. 13499 RRID: AB_2798238 | CUT&Tag (1:100) |
| Antibody | Anti- H2A-K119-ubiquitinylation (rabbit monoclonal) | Cell Signalling Technology | Cat. No. 8240 RRID: AB_10891618 | CUT&Tag (1:100) IFF (1:100) |
| Antibody | Anti-H3-K9-dimethyl (mouse monoclonal) | EMD Millipore | Cat. No. 05–1249 RRID: AB_11210998 | CUT&Tag (1:100) |
| Antibody | Anti-H3-K27-trimethyl (rabbit monoclonal) | Cell Signalling Technology | Cat. No. 9733 RRID: AB_2616029 | CUT&Tag (1:100) |
| Antibody | Anti-H4-K16-acetyl (rabbit monoclonal) | Abcam | Cat. No. ab109463 RRID: AB_10858987 | CUT&Tag (1:100) |
| Antibody | ab5821 anti-fibrillarin (rabbit polyclonal) | Abcam | Cat. No. ab5821 RRID: AB_2105785 | IFF (1:100) |
| Genetic reagent (*Drosophila melanogaster*) | $w^{1118}$ | Bloomington Drosophila Stock Center | BDSC:3605; RRID:BDSC_3605 | |
| Genetic reagent (*D. melanogaster*) | $bam^{D86}$ | Bloomington Drosophila Stock Center | BDSC:5427; RRID:BDSC_5427 | |
| Genetic reagent (*D. melanogaster*) | Df(3R)FDD-0089346 | Bloomington Drosophila Stock Center | BDSC:27402; RRID:BDSC_27402 | Deficiency including *bam* |
| Genetic reagent (*D. melanogaster*) | $aly^1$ | Bloomington Drosophila Stock Center | BDSC:1148; RRID:BDSC_1148 | |
| Genetic reagent (*D. melanogaster*) | Df(3L)BSC428 | Bloomington Drosophila Stock Center | BDSC:24932; RRID:BDSC_24932 | Deficiency including *aly* |
| Genetic reagent (*D. melanogaster*) | Heph-GFP | Bloomington Drosophila Stock Center | BDSC:51540; RRID:BDSC_51540 | $P[PTT-GC]heph^{CC00664}$ |
| Genetic reagent (*D. melanogaster*) | bam-GAL4 | Bloomington Drosophila Stock Center | BDSC:80579; RRID:BDSC_80579 | $P[bam-GAL4:VP16,w^+]1$ |
| Genetic reagent (*D. melanogaster*) | UAS-RFP | Bloomington Drosophila Stock Center | BDSC:30556; RRID:BDSC_30556 | $P[UAS-RFP,w^+]2$ |
| Genetic reagent (*D. melanogaster*) | Pc-GFP | Bloomington Drosophila Stock Center | BDSC:9593; RRID:BDSC_9593 | $P[Pc-eGFP,w^+]3$ |
| Biological sample (*D. melanogaster*) | Wing imaginal discs | This paper | Dissected tissue | |
| Biological sample (*D. melanogaster*) | Adult testes | This paper | Dissected tissue | |
| Biological sample (*D. melanogaster*) | Spermatocytes | This paper | Dissected tissue | |

## Fly strains

All crosses were performed at 25°C. All mutations and chromosomal rearrangements used here are described in Flybase (http://www.flybase.org). The $w^{1118}$ strain was used as a wildtype control. The *heph-GFP* males used for profiling have the genotype *y w/Y; P[PTT-GC]heph$^{CC00664}$/TM3, Ser Sb*. The *bam* mutant males have the genotype *w/Y; e bam$^{D86}$/Df(3R)FDD-0089346*. The *aly* mutant males have the genotype *P[ry11]ry2, mwh aly$^1$ ry$^{506}$ e/Df(3L)BSC428*. Additional genotypes used for cytological characterization were *y w P[bam-GAL4:VP16,w$^+$]1/Y; P[UAS-RFP,w$^+$]2/2; P[PTT-GC]heph$^{CC00664}$/3* and *w$^{1118}$/Y; P[Pc-eGFP,w$^+$]3*.

## Antibodies

The following antibodies were used: Epicypher 13-0027 anti-H3-K4-dimethyl, Cell Signalling Technology E1Z3G anti-RNAPII-Serine-2-phosphorylation, Cell Signalling Technology 8240 anti-H2A-K119-ubiquitinylation, EMD Millipore 05-1249 anti-H3-K9-dimethyl, Cell Signalling Technology C36B11 anti-H3-K27-trimethyl, Abcam ab109463 anti-H4-K16-acetyl, and Abcam ab5821 anti-fibrillarin.

## Imaging whole testes

Testes from 1-day-old adult males were dissected and fixed in 4% formaldehyde/PBS with 0.1% Triton-X100 (PBST) for 10 min, stained with 0.5 µg/mL DAPI/PBS, and mounted in 80% glycerol on slides. Testes were imaged by epifluorescence on an EVOS FL Auto 2 inverted microscope (Thermo Fisher Scientific) with a 10× objective.

## Imaging spermatocytes

Testes from third-instar male larvae were prepared as described (*Bonaccorsi et al., 2000*). Briefly, one testis was dissected in a drop of PBS on a Histobond glass slide (VWR 16004-406), squashed gently with a RainX (ITW Global Brands) coated coverslip, then flash-frozen in liquid nitrogen. After popping off the coverslip, the sample was fixed with 4% formaldehyde/PBST for 5 min, and incubated with 0.3% sodium deoxycholate/PBST twice for 20 min each. Samples were incubated with primary antiserum in PBST supplemented with 0.1% bovine serum albumin (BSA) at 4°C overnight, and finally with fluorescently labeled secondary antibodies (1:200 dilution, Jackson ImmunoResearch). Slides were stained with 0.5 µg/mL DAPI/PBS, mounted in 80% glycerol, and imaged by epifluorescence on an EVOS FL Auto 2 inverted microscope (Thermo Fisher Scientific) with a 40× objective. Dissection and immunostaining was typically repeated at least 10 times to confirm results. We also imaged stained spermatocytes from adult testes, but imaging of larval spermatocytes was typically cleaner with less background. Pseudo-colored images were adjusted and composited in Adobe Photoshop and Adobe Illustrator.

## Whole-mount CUT&Tag

To perform CUT&Tag for whole tissues ('whole-mount CUT&Tag'), we dissected 10 testes from 1-day-old adults or 10 imaginal wing discs from third instar larvae in PBS buffer supplemented with cOmplete Protease Inhibitor (Roche 11697498001). Dissected tissues were permeabilized with 0.1% Triton/PBS for 30 min at room temperature, and then manually transferred into the following CUT&Tag solutions sequentially between wells of a glass dissection plate: primary antibody solution (diluted in Wash+ buffer [20 mM HEPES pH 7.5, 150 mM NaCl, 0.5 mM spermidine, 2 mM EDTA, 1% BSA, with cOmplete Protease Inhibitor]) overnight at 4°C, secondary antibody solution (in Wash+ buffer) for 1 hr at room temperature, and then incubated with loaded protein-A-Tn5 (in 300Wash+ buffer [20 mM HEPES pH 7.5, 300 mM NaCl, 0.5 mM spermidine with cOmplete Protease Inhibitor]) for 1 hr. After one wash with 300Wash+ buffer, samples were incubated in 300Wash+ buffer supplemented with 10 mM MgCl$_2$ for 1 hr at 37°C to tagment chromatin. Tissues were then dissociated with collagenase (2 mg/mL, Sigma C9407) in HEPESCA (50 mM HEPES buffer pH 7.5, 360 µM CaCl$_2$) solution at 37°C for 1 hr. We then added SDS to 0.16%, protease K to 0.3 mg/mL, and EDTA to 16 mM and incubated at 58°C for 1 hr, and DNA was purified by phenol:chloroform extraction and ethanol precipitation. Libraries were prepared as described (*Kaya-Okur et al., 2019*; *Kaya-Okur and Henikoff, 2020*), with 14 cycles of PCR. Libraries were sequenced in PE50 mode on the Illumina NextSeq 2000 platform at the Fred Hutchinson Cancer Center Genomics Shared Resource.

## FACS-CUT&Tag of spermatocytes

40 testes were dissected from 1-day-old adult *heph-GFP* males and digested in 200 µL of 2 mg/mL collagenase (Sigma C9407) in HEPESCA solution at 37°C for 1 hr. The sample was then repeatedly pipetted with a P200 pipette tip to dissociate the tissue, then passed through a 35 µM filter with 5 mL collection tube (Corning 352235) on ice. The filter was washed with PBS to bring the total volume of collected filtrate to 1 mL. A Sony MA900 Multi-Application Cell Sorter with a 100 µM nozzle, flow pressure of 2, GFP laser settings of 32%, and FSC = 1 was used for isolating cells. Isolated cells were collected in 1 mL of PBS in 5 mL tubes. Benchtop CUT&Tag was performed on these samples as described (*Kaya-Okur et al., 2019*), and sequenced in PE50 mode.

## Genome mapping

To streamline analysis of repetitive transposons in the fly genome, we used a modified version of the release r6.30 *D. melanogaster* genome for mapping where repetitive sequences are masked out of the genome (http://hgdownload.cse.ucsc.edu/goldenPath/dm6/bigZips/dm6.fa.masked.gz) and with consensus sequences for 128 transposon sequences (https://github.com/bergmanlab/drosophila-transposons/blob/master/misc/D_mel_transposon_sequence_set.fa) appended (*Ashburner et al., 2021*). Paired-end reads were mapped to this assembly using Bowtie2 (using parameters, e.g.: `--end-to-end` --very-sensitive --no-mixed --no-discordant -q --phred33 -I 10 -X 700).

Mapped reads from whole-tissue replicates were merged using samtools-merge and converted to coverage tracks using bedtools-genomecov with options -scale -fs. For profiling FACS-isolated spermatocytes, duplicate reads were removed from each library using Picard-remove duplicates, and then replicates were merged using samtools-merge and converted to coverage tracks using bedtools-genomecov with options -scale -fs. These tracks are hosted at UCSC (https://genome.ucsc.edu/s/jamesanderson12358/analysis230508___UCSC_session_germline_MS) for visualization, and selected regions were exported as PDF files.

## Processing of FCA testis snRNA-seq data

We downloaded snRNA-seq data generated by the Fly Cell Atlas project (*Li et al., 2022*) as a Seurat object linked in supplementary data of *Raz et al., 2023*, summarizing gene expression data of single nuclei from dissociated *Drosophila* adult testes. We used the Seurat function AverageExpression() to get the average expression of all genes in each of 40 UMAP groups which represent distinct cell types of the testis. This produced a 40 groups × 15,833 genes table.

The 18 germline groups and 22 somatic groups assigned in *Raz et al., 2023*, are:

1. spermatogonium
2. spermatogonium-spermatocyte transition
3. mid-late proliferating spermatogonia
4. spermatocyte 0
5. spermatocyte 1
6. spermatocyte 2
7. spermatocyte 3
8. spermatocyte 4
9. spermatocyte 5
10. spermatocyte 6
11. spermatocyte 7a
12. maturing primary spermatocyte
13. spermatocyte
14. late primary spermatocyte
15. early elongation stage spermatid
16. early-mid elongation-stage spermatid
17. mid-late elongation-stage spermatid
18. spermatid
19. hub
20. cyst stem cell
21. early cyst cell 1
22. early cyst cell 2
23. cyst cell intermediate
24. spermatocyte cyst cell branch a
25. spermatocyte cyst cell branch b
26. cyst cell branch a
27. cyst cell branch b
28. male gonad associated epithelium
29. seminal vesicle
30. adult tracheocyte
31. muscle cell
32. testis epithelium
33. hemocyte
34. hcc
35. tcc

36. pigment cell
37. adult fat body
38. secretory cell of the male reproductive tract
39. adult neuron
40. 'unannotated'

## Promoter and gene scoring tables

We compiled a list of genes with male-germline-enriched expression as follows. We compiled a list of unique protein-coding mRNAs and lncRNA genes from the *Drosophila* dmel_r6.31 genome assembly (http://ftp.flybase.net/releases/FB2019_06/dmel_r6.31/gtf/), and matched FCA expression data for 40 testis cell types (*Raz et al., 2023*) to each gene with a lookup table. There were 1062 genes that are not represented in the expression dataset; these genes are listed with #N/A values for gene expression.

For each gene, we calculated its average expression in the 18 germline groups (gexp) and its average expression in 21 somatic groups (sexp). The 40th 'unannotated' group was not considered for gexp or sexp values. We then used k-means clustering (k=10) to group genes by cell-type expression within the testis (*Figure 2—figure supplement 1*). The k-means groups 1–5 were associated with gene expression in germline clusters 1–18. The remaining groups 6–10 were associated with gene expression in somatic clusters 19–40, and we collapsed these into one somatic group called 'all somatic categories' (k-cluster group 11). We added these k-cluster annotations to each gene.

We then selected 6419 genes with log$_2$fold-change≥1 average expression in germline groups than in testis somatic groups (termed genes with germline-enriched expression). This table is included as *Supplementary file 1c*. To assign alternative promoters to each gene, for each transcript in the .gtf file with orientation '+' we assigned the minimum coordinate as its TSS position, and for each transcript with orientation '-' we assigned the maximum coordinate as its TSS position. We retained only one instance of duplicate TSSs for genes with TSS coordinates represented multiple times. In 144 instances two gene names share the same TSS coordinate, and we retained a TSS for each gene in the table. This table of 21,982 promoters is included as *Supplementary file 1b*.

For each gene, we determined the identity and distance to the nearest promoter of the next upstream gene using the bedtools/closest with parameters: -D a -fu, and for the next downstream gene with parameters: -D a -fd.

To summarize the enrichment of chromatin marks at promoters, we counted mapped reads in an interval from –200 to +500 bp around each TSS in merged profiling data by summing reads using deeptools/multiBamSummary with parameters: BED-file –BED and scaled counts by the number of reads in each library (counts * 1,000,000)/(number of reads in library) to give counts per million. Profiling counts were transformed into z-scores for each promoter between *bam*, *aly*, and wildtype testis samples, and these values are appended to *Supplementary file 1b*.

To summarize the enrichment of chromatin marks across genes, we counted mapped reads from the start to the end of each gene in merged profiling data by summing reads using deeptools/multiBamSummary with parameters: BED-file –BED and scaled counts by the number of reads in each library and the length of the gene (counts * 1,000,000)/(number of reads in library * gene length in kb) to give counts per kilobase per million (CPKM), and these values are appended to *Supplementary file 1c*.

To summarize enrichment of chromatin marks across transposons, we counted mapped reads across consensus transposon sequences using deeptools/multiBamSummary with parameters: BED-file –BED and scaled counts as CPKM. These values are provided in *Supplementary file 1d*.

## Genomic display

For average plots of H3K4me2 signal around promoters, profiling coverage was summarized with deepTools/bamCoverage ± 1 kb around annotated TSSs excluding regions with a second gene promoter in the display window with 10 bp binning, and plotted using plotHeatmap.

For average plots of H3K4me2 signal around promoters, profiling coverage was summarized with deepTools/bamCoverage ± 1 kb around annotated TSSs excluding regions with a second gene promoter in the display window with 10 bp binning, and plotted using plotHeatmap.

For heatmapping of RNAPIIS2p signal at gene starts, we used deepTools/computeMatrix with parameters: -b 2000 -a 2000 -R and then we used deepTools/plotHeatmap with parameters: --colorMap viridis.

For visualizing the chromosomal distribution of CUT&Tag data as CIRCOS plots, we used the circlize package in R with default settings (circlize version 0.4.15). Genome coverage files for plotting were generated by deepTools/bamCoverage command, using parameters:

    -bs 20000 --centerReads --effectiveGenomeSize 142573017 -of bedgraph

And plotted in consecutive 20 kb bins. The innermost three rings of each plot display genomic coverage, with color-coding set independently per ring. The color of bins in the outer ring correspond to the fold-change of signal in spermatocytes compared to wing imaginal discs calculated using deep-Tools/multiBigwigSummary command, with parameters:

    bins -bs 20000 –outRawCounts

For boxplots, the enrichment score for each gene was scaled by dividing its read count by the median count in the 'chromosome *2 & 3*' category and plotted, discarding genes with enrichment score = 0 in either dissociated testes, in spermatocytes, or in wing imaginal discs.

## Additional information

### Funding

| Funder | Grant reference number | Author |
|---|---|---|
| National Human Genome Research Institute | HG010492 | Steven Henikoff |
| Howard Hughes Medical Institute | Henikoff | Steven Henikoff |

The funders had no role in study design, data collection and interpretation, or the decision to submit the work for publication.

### Author contributions

James T Anderson, Data curation, Software, Formal analysis, Visualization, Writing – original draft, Writing – review and editing; Steven Henikoff, Conceptualization, Supervision, Funding acquisition, Writing – original draft, Writing – review and editing; Kami Ahmad, Conceptualization, Data curation, Formal analysis, Supervision, Funding acquisition, Investigation, Visualization, Writing – original draft, Writing – review and editing

### Author ORCIDs

Steven Henikoff (iD) https://orcid.org/0000-0002-7621-8685
Kami Ahmad (iD) https://orcid.org/0000-0001-8572-6182

Reviewer #1 (Public Review): https://doi.org/10.7554/eLife.89373.3.sa1
Reviewer #2 (Public Review): https://doi.org/10.7554/eLife.89373.3.sa2
Author Response https://doi.org/10.7554/eLife.89373.3.sa3

## Additional files

### Supplementary files

• Supplementary file 1. Samples and sequencing results. (a) Sample IDs and sequencing results. (b) snRNA-seq gene expression scores and H3K4me2 enrichment at *Drosophila* promoters. This lists unique TSSs in the Flybase dm6 r6.31 assembly release and associated gene expression scores (derived from *Raz et al., 2023*) and H3K4me2 signal (in counts per million [CPM]) in a –200 to +500 bp window around each TSS. (c) Enrichment of RNAPIIS2p, H3K9me2, H3K27me3, and uH2A across *Drosophila* genes in dissociated testes, in isolated spermatocytes, and in wing imaginal discs.

This lists unique genes in the Flybase dm6 r6.31 assembly release and associated gene expression scores (derived from *Raz et al., 2023*) and chromatin profiling signals (in counts per kilobase per million [CPKM]) across each gene length. (d) Enrichment of RNAPIIS2p, H3K9me2, H3K27me3, and uH2A across consensus transposon sequences in dissociated testes, in isolated spermatocytes, and in wing imaginal discs. This lists a subset of transposon consensus sequences (https://github.com/bergmanlab/drosophila-transposons/blob/master/misc/D_mel_transposon_sequence_set.fa) and associated chromatin profiling signals (in CPKM) across each consensus length.

- MDAR checklist

## Data availability

Sequencing data have been deposited in GEO under accession code GSE225300.

The following dataset was generated:

| Author(s) | Year | Dataset title | Dataset URL | Database and Identifier |
|---|---|---|---|---|
| Anderson J, Henikoff S, Ahmad K | 2023 | Data from: Chromosome-specific maturation of the epigenome in the *Drosophila* male germline | https://www.ncbi.nlm.nih.gov/geo/query/acc.cgi?acc=GSE225300 | NCBI Gene Expression Omnibus, GSE225300 |

The following previously published dataset was used:

| Author(s) | Year | Dataset title | Dataset URL | Database and Identifier |
|---|---|---|---|---|
| Raz AA, Vida GS, Stern SR, Mahadevaraju S, Fingerhut JM, Viveiros JM, Pal S, Grey JR, Grace MR, Berry CW, Li H, Janssens J, Saelens W, Shao Z, Hu C, Yamashita YM, Przytycka T, Oliver B, Brill JA, Fuller MT | 2023 | Data for: Emergent dynamics of adult stem cell lineages from single nucleus and single cell RNA-Seq of *Drosophila* testes | https://doi.org/10.5061/dryad.m63xsj454 | Dryad Digital Repository, 10.5061/dryad.m63xsj454 |

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
