## [Editor Report · eLife assessment]

Using a variety of methods including mutant analyses, the authors study chromatin structure during spermatogenesis in *Drosophila* and transcriptional profiling in single cells/nuclei. This description of the dramatic changes in chromatin structure during spermatogenesis leads to some new observations, with **convincing** evidence, and it is **useful** for the field.

---

## [Referee Report · Reviewer #1 (Public Review)]

Anderson, Henikoff and Ahmad et al. performed a series of genomics assays to study *Drosophila* spermatogenesis. Their main approaches include (1) Using two different genetic mutants that arrest male germ cell differentiation at distinct stages, bam and aly mutant, they performed CUT&TAG using H3K4me2, a histone modification for active promoters and enhancers; (2) Using FACS sorted pure spermatocytes, they performed CUT&TAG using antibodies against RNA PolII phosphorylated Ser 2, H4K16ac, H3K9me2, H3K27me3, and ubH2AK118. They also compare these chromatin profiling results with the published single-cell and single-nucleus RNA-seq data. Their analyses are across the genome but the major conclusions are about the chromatin features of the sex chromosomes. For example, the X chromosome is lack of dosage compensation as well as inactivation in spermatocytes, while Y chromosome is activated but enriched with ubH2A in spermatocytes. Overall, this work provides high quality epigenome data in testes and in purified germ cells. The analyses are very informative to understand and appreciate the dramatic chromatin structure change during spermatogenesis in *Drosophila*.

---

## [Referee Report · Reviewer #2 (Public Review)]

Anderson et al profiled chromatin features, including active chromatin marks, RNA polymerase II distribution, and histone modifications in the sex chromosomes of spermatogenic cells in *Drosophila*. The experiments and analyses were well done, by a combination of the latest and appropriate methods. They include appropriate numbers of replicates. Results were parsed by comparing them among wildtype and two mutant with different arrest stages in spermatogenesis, as well as in FACS-sorted spermatocytes. The authors profiled larval wing discs as reference-somatic cells, allowing focus on features associated with germ cells; comparisons to testis somatic cells provided further specificity. Results were further refined by categorizing genes of interest based on available single nucleus RNA seq expression profiles. The authors acknowledge that the paper's interpretations are based on subtractive logic using the mutants, but comment that more precise ways of staging would not have yielded sufficient sample for their methods.

The authors documented differences in the distribution of RNAPIIS2p on some genes in germ cells vs somatic cells, the presence of a uH2A body beginning in early spermatocytes, and high levels of uH2A on the Y chromosome with little or none on the X, which is intriguing because uH2A is usually associated with silencing, yet the Y chromosome is active in spermatogenic cells. All of these are new, interesting, and important. Also importantly, the authors' data provide molecular details consistent with lack of MSCI, and lack of dosage compensation of the X chromosome in *Drosophila* spermatocytes.

---

## [Author Response]

The following is the authors’ response to the original reviews.

**Public Reviews:**

**Reviewer #1 (Public Review):**
Anderson, Henikoff, Ahmad et al. performed a series of genomics assays to study *Drosophila spermatogenesis*. Their main approaches include (1) Using two different genetic mutants that arrest male germ cell differentiation at distinct stages, bam and aly mutant, they performed CUT&TAG using H3K4me2, a histone modification for active promoters and enhancers; (2) Using FACS sorted pure spermatocytes, they performed CUT&TAG using antibodies against RNA PolII phosphorylated Ser 2, H4K16ac, H3K9me2, H3K27me3, and ubH2AK118. They also compare these chromatin profiling results with the published single-cell and single-nucleus RNA-seq data. Their analyses are across the genome but the major conclusions are about the chromatin features of the sex chromosomes. For example, the X chromosome is lack of dosage compensation as well as inactivation in spermatocytes, while Y chromosome is activated but enriched with ubH2A in spermatocytes. Overall, this work provides high-quality epigenome data in testes and in purified germ cells. The analyses are very informative to understand and appreciate the dramatic chromatin structure change during spermatogenesis in *Drosophila*. Some new analyses and a few new experiments are suggested here, which hopefully further take advantage of these data sets and make some results more conclusive.Major comments:1. The step-wise accumulation of H3K4me2 in bam, aly and wt testes are interesting. Is it possible to analyse the cis-acting sequences of different groups of genes with distinct H3K4me2 features, in order to examine whether there is any shared motif(s), suggesting common trans-factors that potentially set up the chromatin state for activating gene expression in a sequential manner?

While the histone H3K4me2 mark is low and more widespread at genes active in late spermatocytes and in spermatids (shown in Figure 2C and some examples in Figure 1C-D), we suggest that this may be due to a general decrease in the importance of this modification in late spermatogenesis rather than a specific feature of those genes. We point this out in lines 146-152. This idea is supported by the widespread change in RNAPII distribution in all genes in the germline, shown in Figure 3F and supplementary Figure 2.

1. Pg. 4, line 141-142: "we cannot measure H3K4me2 modification at the bam promoter in bam mutant testes or at the aly promoter in aly mutant testes", what are the allelic features of the bam mutant and aly mutant? Are the molecular features of these mutations preventing the detection of H3K4me2 at the endogenous genes' promoters? Also, the references cited (Chen et al., 2011) and (Laktionov et al., 2018) are not the original research papers where these two mutants were characterized.

We have corrected these citations to the original papers. We clarified in the text that the bamΔ86 allele is a deletion of almost all of the coding sequence (reported in Bopp, D., Horabin, J.I., Lersch, R.A., Cline, T.W., Schedl, P. (1993). Expression of the Sex-lethal gene is controlled at multiple levels during *Drosophila oogenesis*. Development 118(3): 797--812.). The aly1 allele is also a P element-induced mutation; it is not molecularly characterized (it was first described here: Lin, T.Y., Viswanathan, S., Wood, C., Wilson, P.G., Wolf, N., Fuller, M.T. (1996). Coordinate developmental control of the meiotic cell cycle and spermatid differentiation in *Drosophila* males. Development 122(4): 1331--1341.) We noticed a lack of reads for various histone modifications in aly mutants in part of the gene, suggesting that the deletion is limited to the promoter and the first exon. Signal for the H3K4me2 modification is at background levels for the distal portion of aly, suggesting that the deletion inactivates the gene.

1. The original paper that reported the Pc-GFP line and its localization is: Chromosoma 108, 83 (1999).

We now cite the first published description of this marker in the male germline and a more detailed one (lines 291-293).

The Pc-GFP is ubiquitously expressed and almost present in all cell types. In Figure 6B, there is no Pc-GFP signals in bam and aly mutant cells.

We apologize, our labeling of the figure was easily overlooked - the bam and aly genotypes do not carry the PcGFP marker, since we didn’t need it for staging the germline nuclei. We have clarified this in the figure.

According to the Method "one testis was dissected", does it mean that only one testis was prepared for immunostaining and imaging? If so, definitely more samples should be used for a more confident conclusion.

We corrected the text to make it clear that all cytological examinations were repeated at least 10 times (lines 438-439).

Also, why use 3rd instar larval testes instead of adult testes?

Generally, we find that immunostaining of the larval testes is cleaner, and we now mention this in the Methods (lines 439-440). We have immunostained both larval and adult testes for these markers with consistent results.

Finally, it is better to compare fixed tissue and live tissue, as the Pc-GFP signal could be lost during fixation and washing steps. Please refer to the above paper [Chromosoma 108, 83 (1999)] for Pc-GFP in spermatogonial cells and Development 138, 2441-2450 (2011) for Pc-GFP localization in aly mutant.

We are using PcGFP staining for staging with antibody detection of other chromatin features, which requires fixed material, although we have compared PcGFP signal in both live and fixed tissue. We have added the 1999 reference for nuclear staging in the male germline.

1. Ubiquitinylation of histone H2A is typically associated with gene silencing, here it has been hypothesized that ubH2A contributes to the activation of Y chromosome. This conclusion is strenuous, as it entirely depends on correlative results.

We agree that this is a correlation. We cite in the text examples where uH2A is associated with gene activation. We have added a comment to clarify that this is a correlation (lines 318-320), and now present an alternative that uH2A on the Y chromosome may be moderating expression from these highly active genes (lines 405-407).

For example, the lack of co-localization of ubH2A immunostaining and Pc-GFP are not convincing evidence that ubH2A is not resulting from PRC1 dRing activity. It would be a lot stronger conclusion by using genetic tools to show this. For example, if dRing is knocked down (using RNAi driven by a late-stage germline driver such as bam-Gal4) or mutated in spermatocytes (using mitotic clonal analysis), would they detect changes of ubH2A levels?

We have tested multiple constructs to knockdown dRING using the bam-GAL4 driver although we have not reported it in the manuscript. These knockdowns have no effect on uH2A staining in the testis, on motile sperm production, or on male fertility, although these RNAi constructs do produce Polycomb phenotypes when expressed in somatic cells from an en-GAL4 driver. This is the reason why we point out in the text that there are multiple alternative candidates for an H2A ubiquitin ligase in the *Drosophila* genome and that in other species RING1 is not responsible for sex body uH2A in the male germline (lines 394-396).

1. Regarding "X chromosome of males is thought to be upregulated in early germline cells", it has been shown that male-biased genes are deprived on the X chromosome [Science 299:697-700 (2003); Genome Biol 5:R40 (2004); Nature 450:238-241 (2007)], so are the differentiation genes of spermatogenesis [Cell Research 20:763-783 (2010)]. It would be informative to discuss the X chromatin features identified in this work with these previous findings.

We now mention that the *Drosophila* X chromosome is moderately depleted of male germline-expressed genes (lines 362-363).

For example, the lack of RNAPII on X chromosome in spermatocytes could be due to a few differentiation genes expressed in spermatocytes located on the X chromosome.

We show in Figure 3B that there is a minor non-significant reduction in RNAPII on the X chromosome in spermatocytes. This small reduction might be due to the moderate paucity of male germline-expressed genes on this chromosome, but since it is non-significant we have not discussed it.

**Reviewer #2 (Public Review):**
Anderson et al profiled chromatin features, including active chromatin marks, RNA polymerase II distribution, and histone modifications in the sex chromosomes of spermatogenic cells in *Drosophila*. The results are new and the experiments and analyses look well done, including with appropriate numbers of replicates. Results were parsed by comparing them among two arrest mutants and wildtype, as well as in FACS-sorted spermatocytes. The authors also profiled larval wing discs to serve as reference-somatic cells, which allowed them to focus only on features in their testis data that were associated with germ cells. Their results were further refined by categorizing the genes of interest based on available single nucleus RNA seq expression profiles. The authors document interesting phenomena, such as differences in the distribution of RNAPIIS2p on some genes in germ cells vs somatic cells, the presence of a uH2A body beginning in early spermatocytes, and high levels of uH2A on the Y chromosome and little or none on the X. The former is intriguing because this modification is usually associated with silencing, yet the Y chromosome is active in spermatogenic cells. The authors interpret some of their data as implying a lack of dosage compensation of the X chromosome in spermatocytes.The data are believable and new, but it is not fully clear how to interpret them. The paper's interpretations rely on subtractive logic to parse results from mixtures of cells down to cell type, extracting spermatogonia, spermatocyte, etc. features by comparing bam mutants (only spermatogonia) to aly mutants (spermatogonia and early spermatocytes but no later stages) to wildtype (all spermatogenic stages), and extracting testis germline data by comparison to wing disc soma; their FACS sorted spermatocytes also have heterogeneity. I recognize that the present paper was a lot of work and am not suggesting that the authors redo their study using methods that give more purity and precision of stage (https://doi.org/10.1126/science.aal3096, https://doi.org/10.1101/gad.335331.119), but they should be aware of them and of their results.

The pulse-release system that the reviewer points to is an interesting system, but more limited in material and in useable markers than the systems we used here. We have added to our discussion of the the limitations of subtractive comparisons between arrest genotypes, both in regards to using mutants that may alter gene expression programs, and to how subtractive comparisons may limit our detection of differences between cell types (lines 143-147).

The conclusions about dosage compensation are indirect, but are consistent with the current model documented in the studies cited by the authors, as well as earlier studies (doi: 10.1186/jbiol30).

We disagree; our data directly speaks to the molecular mechanisms at play. Our profiling of the H4K16 acetylation mark and RNAPII in isolated spermatocytes (Figure 4) demonstrates that current models are correct, and so are useful for settling this point in the literature.

**Reviewer #1 (Recommendations For The Authors):**
Throughout the manuscript, it is better to cite the original research papers.

We have added citations for the original characterizations of bam and aly alleles used, for the descriptions of PCGFP in spermatocytes, and for issues raised by reviewer comments.

Minor comments:Pg.2, line 70-71: "Germline stem cells at the apical tip of the testis asymmetrically divide to birth spermatogonia", should be gonialblast.

Fixed (line 71).

Pg.2, line 71: "four rapid mitotic divisions", the spermatogonial cell cycle lasts several hours-- "rapid" is subjective and relative, better to leave this word out.

Fixed (line 71).

**Reviewer #2 (Recommendations For The Authors):**
Other than the major issue raised in the public review this paper only needs a few minor modifications, listed by line number below. The first one would be considered essential by this reviewer.27: In the sentence that ends on this line, please add the word testis after *Drosophila*.

Fixed (line 27).

119: It must be known from the Fly Cell Atlas data whether these genes do begin to express in spermatogonia.

Collated expression values from the FCA are provided in Supplementary Table 2. In many cases there is detectable expression of these genes in spermatogonia, although transcript abundance peaks in early spermatocytes.

198: remove "distribution of".

Fixed (line 200).

311: enrichment relative to what?

Fixed (line 313). It is relative to signal in wing discs.

344: other aspects could be regulated such as elongation, termination.

We have added caveats to our speculations in this sentence (lines 340-356). The increased signal we see in gene bodies could be due to slower RNAPII elongation, but we don’t see a way that changes in termination would produce this pattern.

369: This part of the paper seems overly speculative, given the many molecular differences between dosage compensation mechanisms of *Drosophila* vs mammals, and studies that indicate that MSCI does occur in *Drosophila* (DOI: 10.3390/genes12111796).

We disagree, and this is a central point in our manuscript. The paper referred to here does not directly assess MSCI in *Drosophila*, instead they argue that MSCI could be the force driving the evolutionary depletion of male-germline-expressed genes they describe. These and many studies in the literature have conflated the effects of a lack of X dosage compensation and of MSCI in the male germline. Our direct measurements of RNAPII in spermatocytes demonstrates that there is no dosage compensation nor is there MSCI. Further, profiling of histone modifications associated with Drosophila somatic dosage compensation (H4K16ac) or with mammalian MSCI (uH2A, H3K9me2) show that the molecular mechanisms found in these other settings are not in play in the *Drosophila* male germline. As we have established these biological differences between mammals and *Drosophila*, it is appropriate to now speculate on why these differences may be, which we do on lines 374-384.

(several lines): Can the authors justify their assumption that chromatin features of larval wing disc cells will match those of somatic cells of adult testes?

We don’t only compare germline features to somatic cells of the wing disc, but also to genes with somatic expression in the testes annotated by FCA expression data (H3K4me2 in Figure 2C, RNAPII in Figure 3F). Note in Supplementary Figure 2 the distribution of RNAPII in whole testes (which includes somatic cells) is similar to that of larval wing discs, confirming that the differences we describe are specific to germline cells.